# Prediction of pH Value by Multi-Classification in the Weizhou Island Area

**DOI:** 10.3390/s19183875

**Published:** 2019-09-08

**Authors:** Haocai Huang, Rendong Feng, Jiang Zhu, Peiliang Li

**Affiliations:** 1Ocean College, Zhejiang University, Zhoushan 316021, China (H.H.) (R.F.) (J.Z.); 2Laboratory for Marine Geology, Qingdao National Laboratory for Marine Science and Technology, Qingdao 266061, China

**Keywords:** ocean acidification, pH, multi-classification, multiple linear regression, softmax regression, support vector machine

## Abstract

Ocean acidification is changing the chemical environment on which marine life depends. It causes a decrease in seawater pH and changes the water quality parameters of seawater. Changes in water quality parameters may affect pH, a key indicator for assessing ocean acidification. Therefore, it is particularly important to study the correlation between pH and various water quality parameters. In this paper, several water quality parameters with potential correlation with pH are investigated, and multiple linear regression, softmax regression, and support vector machine are used to perform multi-classification. Most importantly, experimental data were collected from Weizhou Island, China. The classification results show that the pH has a strong correlation with salinity, temperature, and dissolved oxygen. The prediction accuracy of the classification is good, and the correlation with dissolved oxygen is the most significant. The prediction accuracies of the three methods for multi-classifiers based on the above three factors reach 87.01%, 87.77%, and 89.04%, respectively.

## 1. Introduction

Human activities such as the use of fossil fuels lead to an increase in atmospheric CO_2_ concentrations. It increased from 28 Pa before the Industrial Revolution to 41 Pa (November 2016), an increase of about 46%, and continues to grow at a rate of 0.5% per year [1]. However, as the concentration of CO_2_ in the atmosphere continues to increase, the amount of CO_2_ absorbed by the ocean increases, causing the alkalinity of the surface seawater to decrease, leading to ocean acidification [2]. Research based on the Coupled Model Intercomparison Project Phase 5 shows that in high-concentration emissions scenarios, the pH and primary production of the global ocean will decrease by 0.330 (±0.003) and 8.6% (±7.9%) by the end of the 21st century, respectively, and the concentration of chlorophyll a, phosphate, and nitrate in the Southern Ocean will be significantly reduced [3,4,5].

An important property of aqueous solutions is pH because it affects chemical and biochemical properties such as chemical reactions, equilibrium conditions, and biological toxicity [6]. The changes in ocean chemistry caused by ocean acidification are changing the chemical environment in which marine life depends. The metabolic processes of marine organisms will be affected and the stability of marine ecology will change [7]. pH is one of the key indicators of ocean acidification. Reasonable monitoring of water parameters and correct prediction of pH can effectively avoid the adverse effects of ocean chemical environment and biological ecology caused by ocean acidification. Therefore, it is important to study the relationship between pH and dynamic water quality parameters that vary significantly in the ocean.

Modeling and predicting the effect of nutrients and other environmental variations on pH have been extensively studied in the current literature. Blackford and Gilbert [8] used a coupled carbonate system–marine ecosystem–hydrodynamic model to simulate the temporal and spatial variability in pH across the southern North Sea and simulated the observed inhibition of pelagic nitrification with decreasing pH. Turner, Pelletier, and Kasper [9] used the water quality model QUAL2Kw to conclude that nutrient enrichment of the South Umpqua River, Oregon, was linked to periphyton growth and large diel fluctuations in dissolved oxygen and hydrogen ion (pH) concentrations. Robert-Baldo and Morris [10] used spectrophotometric methods to determine the pH of seawater, and the relationship between temperature and salinity on the phenol red equilibrium constant was obtained to characterize the relationship between temperature and salinity and pH. Halevy and Bachan [11] developed a statistical model of seawater pH as a function of pCO_2_ (atmospheric partial pressure of CO_2_) and the suite of parameters that govern ocean chemistry. It is noted that all these studies were based on the modeling of continuous pH values. In this paper, the pH is regarded as a discrete quantity, which is regarded as a multi-classification problem.

The goal of this paper is to develop a reliable approach to model the correlation between pH and the water quality parameters, and predict pH by water quality parameters. To achieve this goal, several different water parameters, which have potential correlation with pH, are selected. Multiple linear regression, softmax regression, and multi-classification support vector machine are employed to obtain the correlation between these parameters and pH. We use cross-validation methods to get a reliable and stable model and compare the accuracies of three methods for pH value prediction.

## 2. Materials and Methods

### 2.1. Data Collection

The system is placed on Weizhou Island in Beihai, Guangxi, about 600–800 m offshore and is located at a depth of about 3 m. The latitude and longitude coordinates of the system are N 21°4′57.81″, E 109°7′32.31″. The main observation factors include velocity profile and seabed temperature, salinity, depth, chlorophyll, dissolved oxygen, pH, turbidity, etc. The system is mainly equipped with Norway Nortek flow profiler and American multi-parameter water quality meter AML (seven parameters). The meter is used in Weizhou Island to sample seawater every 60 s in a certain depth range, analyze various parameters of seawater, and send them to the host. In this experiment, seawater quality parameters include salinity (Sal), temperature (T), depth (Dep), dissolved oxygen (DO), chlorophyll a (Chl), and pH. In the experiment, the data were continuously recorded daily from 2 October 2018 to 28 November 2018 and 78,701 sets of data were recorded. Since the pH changes very little in seawater and the accuracy of the instrument is not high enough, only five different initial values can be measured which are 2.13, 2.14, 2.15, 2.16, and 2.17. According to the conversion formula pH = initial value × (−9.465) + 28.383, five different pH values can be obtained, namely 7.8439, 7.9386, 8.0332, 8.1279, 8.2226, and the results are just applied to the multi-classification problem.

### 2.2. Problem Set up

Consider the problem as follows: The matrix X∈ℝM×N, *M* = 5, looking for a mapping relationship: f(X)→y, which can make **X** accurately map to the corresponding **y**. Here, the matrix **X** and **y** are of the form:(1)X=[xSalxTxDepxDOxChl]=[xSal(1)xSal(2)xSal(3)…xSal(N)xT(1)xT(2)xT(3)…xT(N)xDep(1)xDep(2)xDep(3)…xDep(N)xDO(1)xDO(2)xDO(3)…xDO(N)xChl(1)xChl(2)xChl(3)…xChl(N)],
(2)y=[y(1)y(2)y(3)…y(N)],
where **X** is the water parameter matrix containing salinity (Sal), temperature (T), depth (Dep), dissolved oxygen (DO), and chlorophyll a (Chl). **y** is the matrix vector of pH. x(⋅)(i) denotes the *i*th value of x(⋅). y(i) denotes the *i*th value of **y**. In the measurement process, the pH has 5 different values, thus y(i)∈{pH1,pH2,pH3,pH4,pH5}.

### 2.3. Data Pre-Processing

In the actual training process, due to the uncertain factors in the seawater and the instrument itself, the fluctuation of the factor of Chl is obviously too large, which has a serious interference effect on the establishment of the prediction model. The chlorophyll content in seawater is unevenly distributed, and the sampling interval of the instrument is one minute. When the instrument detects the chlorophyll-rich seawater brought by the current, it produces outliers. The BoxPlot [12] uses Inter Quartile Range (IQR) for efficient identification of outlier values to prevent these values to influence the quality of outcomes. Compared with the 3-Sigma [13] method or Z-score [14] method based on normal distribution, it is not affected by outliers and can accurately and stably depict the discrete distribution of data. Removing outliers detected by box plot makes the chlorophyll a concentration more statistically significant. The chlorophyll a concentration before and after treatment is shown in Figure 1.

Data should be pre-processed by standardization before multi-classification to reduce the order of magnitude difference between different data. In the field observation data, each water quality parameter has different dimensions and dimension units, which will affect the results of the data analysis. In order to eliminate the dimensional influence between indicators, data standardization processing is needed to resolve the comparability between data indicators, so that the pre-processed data is limited to a certain range (such as [0,1] or [−1,1]), thereby eliminating the adverse effects caused by singular sample data. The following two methods are predominantly used.

#### 2.3.1. Min-Max Normalization

The method of min-max normalization [15] is a linear transformation of the original data, mapping the resulting values between [0 1]. The conversion function is:(3)$$x_{k}=\left(x_{k}-x_{\min}\right)/\left(x_{\max}-x_{\min}\right),$$
where $x_{\max}$ and $x_{\min}$ are the maximum and minimum values of $x_{\mathrm{Sal}}$, and $x_{k}$ is the pre-processed data. We normalize other parameter vectors, including $x_{T},x_{\mathrm{Dep}},x_{\mathrm{DO}},x_{\mathrm{Chl}}$.

#### 2.3.2. Z-Score Standardization

The method of z-score standardization [16] gives the mean and the standard deviation of the raw data for standardization of the data. It applies to situations where the maximum and minimum values of the data are unknown, or where there is outlier data outside the range of values. The processed data conforms to the standard normal distribution, i.e., the mean is 0, the standard deviation is 1, and the conversion function is:(4)$$x_{k}=\left(x_{k}-x_{\mathrm{mean}}\right)/x_{\mathrm{var}},$$
(5)$$x_{\mathrm{var}}=\sqrt{\frac{1}{N}\sum_{i=1}^{N}{\left(x_{i}-\mu \right)}^{2}},$$
where $x_{\mathrm{mean}}$ and $x_{\mathrm{var}}$ are the mean and variances of $x_{\mathrm{Sal}}$, and $x_{k}$ is the pre-processed data. We normalize other parameter vectors, including $x_{T},x_{\mathrm{Dep}},x_{\mathrm{DO}},x_{\mathrm{Chl}}$.

For multi-class support vector machines, after uneven scaling in each dimension, the optimal solution is not equivalent to the original. It must be standardized so that the model parameters are not disturbed by data with a large or small distribution range, and the speed of the gradient can be accelerated to find the optimal solution [17]. In the establishment of mathematical models, it was found that the use of min-max standardization performed better in multi-class Support Vector Machine (SVM), while the Z-score standardization method performed better in softmax regression and multiple linear regression.

### 2.4. Multi-Classification Methods

#### 2.4.1. Multiple Linear Regression (MLR)

MLR is a traditional method mainly used to cope with the multiparameter problems [18]. MLR models the relationship between dependent variable and two or more independent variables by linear regression and makes predictions about new data. The MLR function is given by
(6)$$h_{\theta }(X)=\theta_{0}+\overset{M}{\underset{j=1}{\sum }}\theta_{j}X_{j}=\theta^{T}X_{j},$$
where $h_{\theta }(X)$ is the prediction value of **y**, $X_{j}$ is the independent variable, $\theta_{i}$ is the regression coefficient, and $\theta_{0}$ is the value of the intercept in the linear fitting. pH can be expressed like this:(7)$$h_{\theta }(X)=\theta_{0}+\theta_{1}x_{\mathrm{Sal}}+\theta_{2}x_{T}+\theta_{3}x_{\mathrm{Dep}}+\theta_{4}x_{\mathrm{DO}}+\theta_{5}x_{\mathrm{Chl}},$$
where $x_{\mathrm{Sal}},x_{T},x_{\mathrm{Dep}},x_{\mathrm{DO}},x_{\mathrm{Chl}}$ represent salinity, temperature, depth, dissolved oxygen, and chlorophyll a, respectively. The cost function of MLR is defined as:(8)$$J(\theta )=\frac{1}{2N}{\sum_{i=1}^{N}\left(h_{\theta }(X^{\left(i\right)})-y^{\left(i\right)}\right)}^{2}.$$

pH has only 5 determined values, and then multi-classification is performed according to the distance between the test points and the 5 values. The minimum value of the distance from the test point is the pH of the test point.

#### 2.4.2. Softmax Regression

The softmax regression model [19] is another algorithm for solving multi-class regression problems and is a classifier widely used in the deep network supervised learning part of current deep learning research [20]. Softmax regression is a supervised learning algorithm that requires both input x and the corresponding expected output y. The experimental data {(X(1),y(1)),(X(2),y(2)),⋯,(X(N),y(N))} will be obtained, X(i)∈ℝ6,i=1,2,⋯,N denotes the *i*th column of **X** and contains 6 components, the intercept term, salinity (Sal), temperature (T), depth (Dep), dissolved oxygen (DO), and chlorophyll a (Chl), i.e., X(i)=[1 xSal(i) xT(i) xDep(i) xDo(i)xChl(i)]T. y(i)∈{1,2,3,4,5}, corresponds to pH_1_, pH_2_, pH_3_, pH_4_, pH_5_.

There are 5 different pH values in total, corresponding to k = 5 classifications. For each input X(i) there will be a probability p(y=j | X) for each class. From a vector perspective, it is:(9)hθ(X(i))=[p(y(i)=1|X(i);θ)p(y(i)=2|X(i);θ)⋮p(y(i)=k|X(i);θ)]=1∑j=1keθT⋅X(i)[eθ1TX(i)eθ2TX(i)⋮eθkTX(i)],
where θ1,θ2,…,θk∈ℝen+1 is the parameters of the model.

The cost function of softmax regression [21] is defined as follows:(10)$$\begin{array}{l} J(\theta )=-\frac{1}{m}\left[\sum_{i=1}^{m}\sum_{j=1}^{k}1\left\{y^{\left(i\right)}=j\right\}\cdot \log\left(p\left(y^{\left(i\right)}=j|X^{\left(i\right)};\theta \right)\right)\right]\\ +\frac{\lambda }{2}\sum_{i=1}^{k}\sum_{j=0}^{n}\theta_{ij}^{2} \end{array}$$
(11)$$p\left(y^{\left(i\right)}=j|X^{\left(i\right)};\theta \right)=\frac{\exp\left(\theta_{i}^{T}X^{\left(i\right)}\right)}{\sum_{k=1}^{k}\exp\left(\theta_{k}^{T}X^{\left(i\right)}\right)},$$
where 1{·} is the indication function. If the expression in curly braces is true, the function is indicated as 1; otherwise, it is 0, and $\frac{\lambda }{2}\sum_{i=1}^{k}\sum_{j=0}^{n}\theta_{ij}^{2}$ is a penalty term.

The cost function is generally minimized by a gradient descent optimization algorithm, and the coefficients $\theta_{1},\theta_{2},\theta_{3},\theta_{4},\theta_{5}$ are obtained. In the softmax classifier, when classifying an unknown sample, the corresponding vector is tested separately on the classifier, and the size of the result is used to characterize the probability of belonging to the class. We can know the probability of belonging to each class and think that the probability is unknown. The sample belongs to the class with the highest probability.

#### 2.4.3. Multi-Class Support Vector Machine

Support Vector Machine (SVM) is widely used as a typical classification algorithm [22]. Support Vector Machine (SVM) is a generalized linear classifier for binary classification of data in supervised learning. As a two-class classifier, it can only distinguish between two different pHs. Multi-classification of support vector machines currently uses two methods of all-together and combined coding [23,24]. The all-together method is to modify the objective function and strives to solve all classification problems at one time. However, because of its high computational complexity, the implementation is difficult, and the accuracy is not superior to the indirect method, so it is not commonly used. In this paper, the combination coding method is mainly used to construct multi-class classification by constructing multiple two-class SVMs and combining coding methods.

The combined coding method includes one-against-all method [25], one-against-one method [26], error correction output coding method [27], and minimum output coding method. Hsu and Lin [23] concluded that no method can compete with the one-against-one method in the training time and no method is statistically better than the method in the generalization performance.

One-against-all method is to design a hyperplane between any two classes of samples, so k classes of samples need to design *k*(*k* − 1)/2 classifiers. When classifying an unknown sample, the corresponding vector is tested on the classifier. Each classifier gives one vote to the winning class. The class with the most votes is the class of the unknown sample.

Through experiments, we have five different pH values, so we need to build 10 two-class support vector machines. An example of a classifier is created by pH_1_ and pH_2_. The pre-processed data {(X(1),y(1)),(X(2),y(2)),⋯,(X(N),y(N))} is a training set of *N* = 78105. X(i)∈ℝ6,i=1,2,⋯,N, i.e., X(i)=[1 xSal(i) xT(i) xDep(i) xDo(i)xChl(i)]T, and y(i)∈{1,−1} corresponds to pH_1_, pH_2_. Then there is an optimal classification hyperplane that satisfies the following conditions. The hyperplane makes the point-to-plane distance of any sample greater than or equal to 1.
(12){wTX(i)+b≥1, y(i)=1wTX(i)+b≤−1, y(i)=−1,
where w, b are the normal vectors and intercepts of the hyperplane.

In Figure 2, all samples above the upper supported plane belong to the positive class, and samples below the lower supported plane belong to the negative class. The distance between two supported planes d=2/‖w‖ is defined as the margin, and the positive and negative samples at the supported plane are support vectors. SVM is to get the hyperplane that can correctly divide the data and maximize the margin. The problem can be expressed in the following mathematical expressions:(13)$$\begin{array}{cc} \max_{w,b} & \frac{2}{\left\|w\right\|}\\ s.t. & y^{\left(i\right)}\left(w^{T}X^{\left(i\right)}+b\right)-1\geq 0 \end{array}.$$

Obviously, maximizing d=2/‖w‖ is equivalent to minimizing ‖w‖. For the convenience of calculation, Formula (9) is transformed into the following:(14)$$\begin{array}{cc} \min_{w,b} & \frac{1}{2}\|w{\|}^{2}\\ s.t. & y^{\left(i\right)}\left(w^{T}X^{\left(i\right)}+b\right)-1\geq 0 \end{array}.$$

There is a case where the dimension of the feature space is large. Therefore, one constructs the Lagrangian.
(15)$$\begin{array}{l} \min_{\alpha }\frac{1}{2}\sum_{i=1}^{N}\sum_{j=1}^{N}\alpha_{i}\alpha_{j}y^{{}_{\left(i\right)}}y^{{}_{\left(j\right)}}K\left(x^{{}_{\left(i\right)}},x^{{}_{\left(j\right)}}\right)-\sum_{i=1}^{N}\alpha_{i}\\ s.t.\;\sum_{i=1}^{N}\alpha_{i}y^{{}_{\left(i\right)}}=0\\ 0\leq \alpha_{i}\leq C,\;i=1,2,\cdots ,N \end{array},$$
where $\alpha_{i}$ is Lagrange multiplier, $K\left(x_{i},x_{j}\right)$ is the kernel function, and *C* is the soft edge penalty parameter, which is used to balance the training error and generalization ability of the algorithm. According to Mercer’s Theorem, the linear kernel function is used here.
(16)$$K\left(x,x^{\prime }\right)=x^{T}x^{\prime }$$

In the same way, the remaining nine two-class classifiers are built. When an unknown sample is classified and its pH is predicted by one-against-one, the corresponding vector is tested on the classifier, and then the voting form is adopted. The class is the pH of the unknown sample, and the multi-classification based on the support vector machine can be completed.

#### 2.4.4. Summary

Multiple linear regression is used to fit the curve, and the multi-classification is performed by the distance. The cost function of multiple linear regression ignores the decision function of judging distance. Therefore, the prediction accuracy is not high. Softmax regression can get the possibility of the sample for each category and can show the accuracy of different classifications. The SVM can only get a certain result, and the deviation between the predicted result and the actual result cannot be obtained to indicate the accuracy of the prediction.

## 3. Experimental Results

A total of 78,701 data sets were used, and these data were divided into five groups according to the difference in pH. By using the 10-fold cross-validation method, each group was divided into 10 equal parts, and then nine parts were taken in each group as the training model, and the rest of the data was used as the test group. The 10 times 10-fold cross validation was used to obtain the average value of the prediction accuracies for each pH is obtained as an estimate of the accuracies of the methods. The data volume of the five groups of pH is about 1:13:210:160:7.

Table 1 shows the mean and root mean squared error (RMSE) of each coefficient by the MLR model before and after data pre-processing. Obviously, the value of the regression coefficient does not change much after the data processing, but the RMSE has been significantly reduced. The reduction in RMSE illustrates that data processing enhances the reliability of the model. The classifier is suitable for this research goal of predicting pH through water body parameters.

Figure 3 and Figure 4 are the error bar graphs of multi-classifier coefficients by softmax regression before and after data pre-processing, which characterizes the mean and root mean squared error (RMSE) of the classifier coefficients under 10 subgroups. Figure 5 and Figure 6 are the error bar graphs of multi-classifier coefficients by SVM before and after data pre-processing. When softmax regression and support vector machines are used to multi-classify pH, the fluctuation values of their classifier coefficients are also very small after data pre-processing, which improves the overall performance of the multi-classifier. Then it is initially recognized that the absolute value of the coefficient of the DO factor is still large in both the SVM classifier and the softmax regression, and there is a great correlation with the prediction of pH.

Figure 7 and Figure 8 show the training accuracies and test accuracies of the three multi-class methods before and after data pre-processes, respectively. The training accuracy and test accuracy are respectively the ratio of the quantity predicted by the model to the actual result and the total quantity, in the training set and the test set. It can be found that the accuracies of the three models are very similar, and the test accuracy is slightly lower than the training accuracy, indicating that the model has no overfitting and the pH prediction is reliable. After the data is preprocessed, the accuracies of pH_1_, pH_2_, pH_3_, and pH_4_ have been improved accordingly, and pH_5_ has decreased, but the overall accuracy is improved. The data preprocessing has a significant improvement on the reliability of the model. The training accuracies and test accuracies by MLR and softmax regression for pH_1_ are almost zero. This is mainly because when the cost function is calculated, the data volume of pH_1_ is too small, and the weight in the cost function is too small. In contrast, the multi-classifier by SVM performs very well at pH_1_. Overall, the SVM is superior to the other two methods in predicting the pH. In addition, the RMSE of the accuracy is within acceptable limits.

To simplify the comparison between the different models, we calculated the mean and RMSE of the prediction accuracy. Considering that not all five dependent variables have a strong correlation to pH prediction. Therefore, the prediction accuracy is taken as a reference index for the reliability of the model. The multi-classification was performed by removing the single dependent variable or by retaining the single dependent variable. Before being processed, the prediction accuracies of the classifier based on MLR, softmax regression, and SVM are 87.14%, 87.96%, and 89.11%, respectively, as a measure of the change in prediction values before and after processing.

Figure 9 shows the mean and RSME of the prediction accuracies by three methods. Overall, MLR has the lowest prediction accuracy and SVM has the highest prediction accuracy. It is worth noting that when the dependent variable DO is not considered, it is found that the prediction accuracies of the three methods are greatly reduced. This greatly explains that the DO has an important relationship with the prediction of pH. When the Chl and Dep are removed separately, it is found that the average values of the prediction accuracies of the three methods is not changed much and the variation range is not more than 0.2%. It is considered that within a certain range, the effect of both on pH prediction can be ignored. When the two dependent variables Sal and T are not considered, the prediction accuracy of MLR decreases from 87.14% to 83.84% and 86.23%, respectively. The prediction accuracy of softmax regression decreases from 87.96% to 85.75% and 86.98%, respectively. The prediction accuracy of SVM decreases from 89.11% to 86.99% and 87.85%, respectively. Sal and T have a certain effect on the pH prediction, but the effect is much less than DO.

After screening single factors and performing single factor multi-classification, it can be found that the prediction accuracies of classifiers based on Sal, T, or DO are higher than 60%. Among them, the prediction accuracies of classifier based on DO is as high as 72.45%, 75.03%, and 75.55% in the three multi-classification methods, which proves that DO has a great influence on the pH prediction. The prediction accuracy of the single factor classifiers based on Sal or T is second. The changes in the prediction accuracy after removing and retaining factors are just the opposite, indicating the broad applicability of the conclusions. Among the five factors, the three factors of DO, Sal, and T have the highest correlation with pH prediction, and Dep and Chl have almost no effect. In order to simplify the multi-classification model for predicting pH, the two poorly correlated factors, Dep and Chl, are not considered, and the model of the classifier is further simplified in different ways. Then, the final method of simplifying the prediction model is determined by observing the change in the prediction accuracy.

To simplify the model, we reduce the complexity of the model by reducing the dependent variable, making the model easier to calculate. From Figure 10, Dep and Chl have little correlation with pH. When the two factors of Dep and Chl are removed, there is no significant change in the average of the prediction accuracy. After parameters Dep and Chl are removed, parameters Sal and T are removed in different orders. When parameter Sal is removed first, there is a great change on prediction accuracy and it drops by 3%. However, while after removing parameter T first, the prediction accuracy only drops by 1%. When they are all removed, the prediction accuracy drops significantly, dropping more than 13%. According to the change of the prediction accuracy, the three factors of Sal, T, and DO are used to establish a multi-classification model. The prediction accuracies of multi-classifiers using three factors in three ways are 87.01%, 87.77%, and 89.04%. Compared with the original correct rate, it only reduced by about 0.1%, but the model is greatly simplified.

The relative stability of seawater pH is primarily controlled by the dissociation equilibrium of the dissolved inorganic carbonate. Marine plants absorb free carbon dioxide during photosynthesis, while releasing the same amount of oxygen, thus reducing the pH of seawater. The pH of seawater will decrease slightly with increasing temperature, which is the result of the increase in the ionization constant of weak acid in seawater with increasing temperature. As the seawater salinity increases, the ionic strength will increase, and the degree of ionization of carbonic acid in seawater is affected. Thereby, the activity coefficient and activity of the hydrogen ions increases, that is, the pH of the seawater decreases. As mentioned above, this and the results obtained after modeling are mutually validated, which further verifies the reliability of the model.

## 4. Conclusion

The massive discharge of CO_2_ gas leads to acidification of the ocean, affecting the water quality parameters and pH of the ocean. In this paper, MLR, softmax regression, and support vector machine were used to predict the correlation between selected water quality parameters and pH value by multi-classification. A total of 78,107 data sets were used in the experiment, and 10-fold cross-validation methods were used.

The mean and RMSE of the prediction accuracies were computed, which were major criteria used to evaluate the performances of different methods. The results show that the multi-classifier based on support vector machine has higher prediction accuracy than MLR and softmax regression. SVM can also make good predictions for pH with less data volume, while the other two do not. Moreover, the model can be simplified by removing the two factors Chl and Dep. The prediction accuracies of the three-factor multi-classification model of DO, T, and Sal are almost not reduced compared with five factors, and the prediction of pH is the most relevant to dissolved oxygen. Another goal in the future is to consider more water parameters, such as transparency, in related testing and modeling.

## Figures and Tables

**Figure 1 sensors-19-03875-f001:**
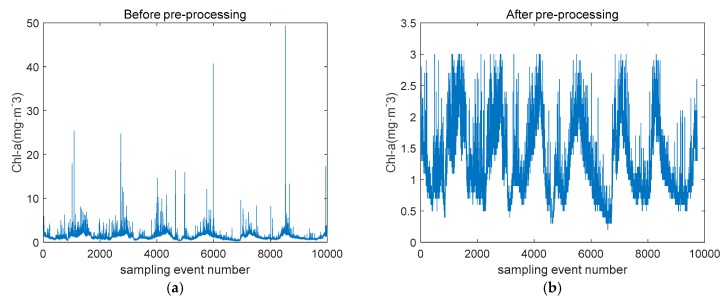
Data pre-processing results.

**Figure 2 sensors-19-03875-f002:**
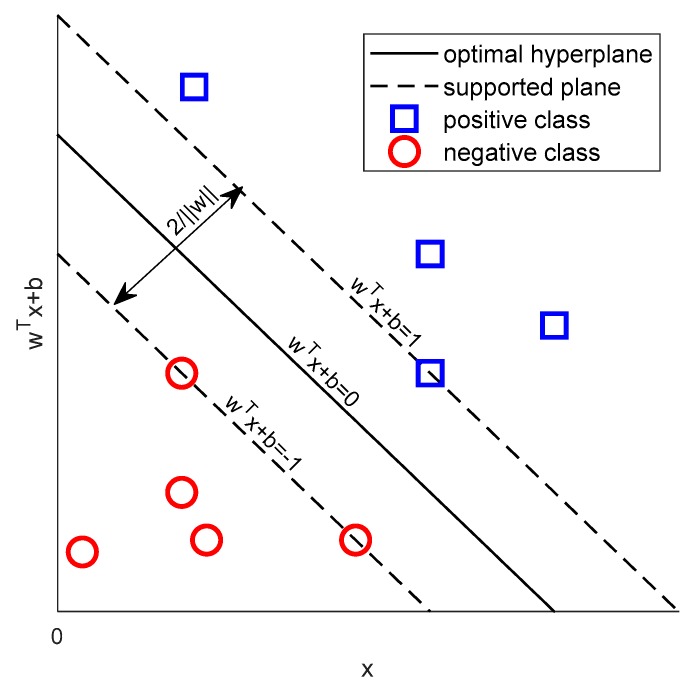
Hyperplane schematic diagram.

**Figure 3 sensors-19-03875-f003:**
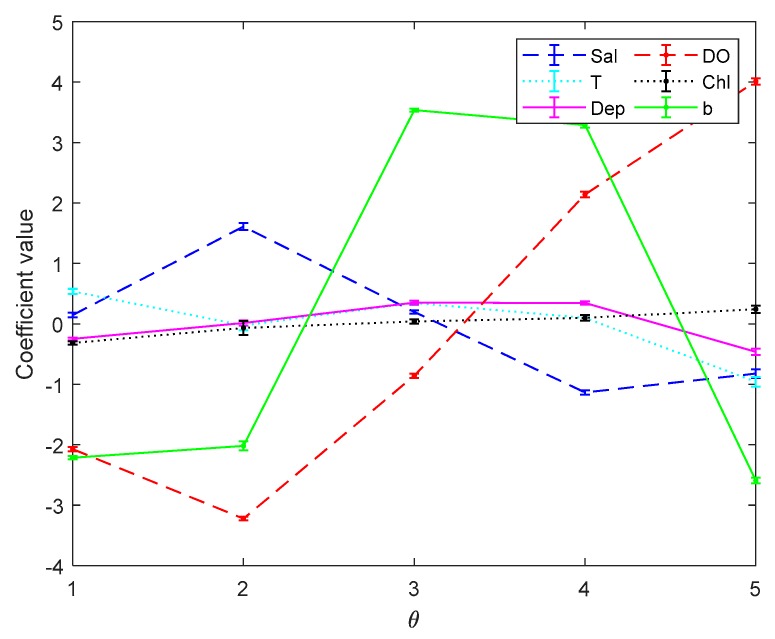
Coefficients by softmax regression before data preprocessing.

**Figure 4 sensors-19-03875-f004:**
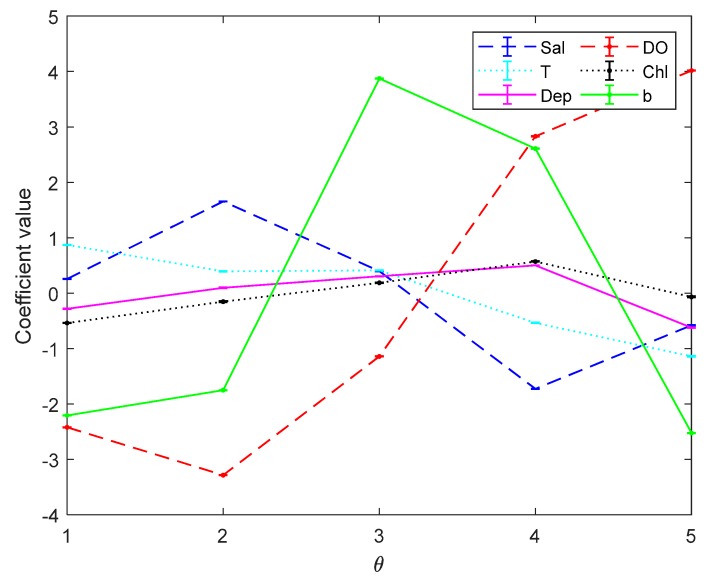
Classifiers by softmax regression after data preprocessing.

**Figure 5 sensors-19-03875-f005:**
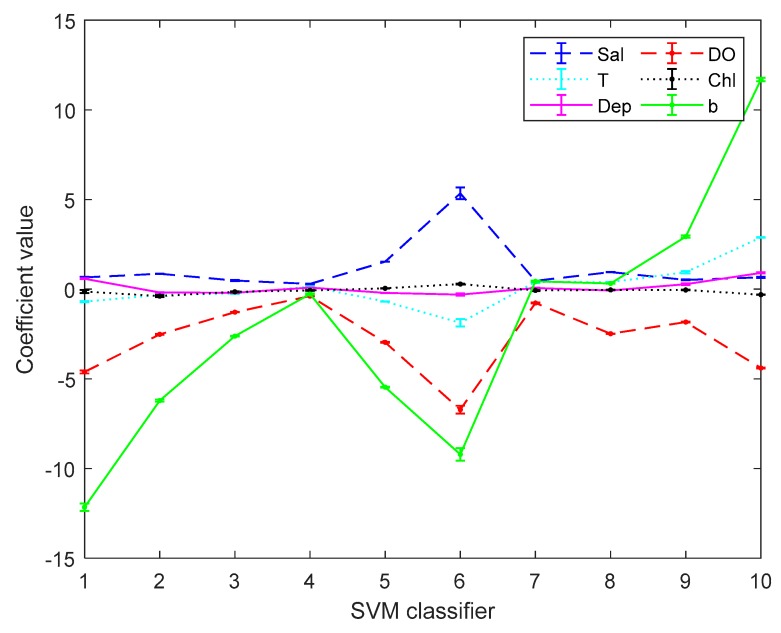
Coefficients by Support Vector Machine (SVM) before data preprocessing.

**Figure 6 sensors-19-03875-f006:**
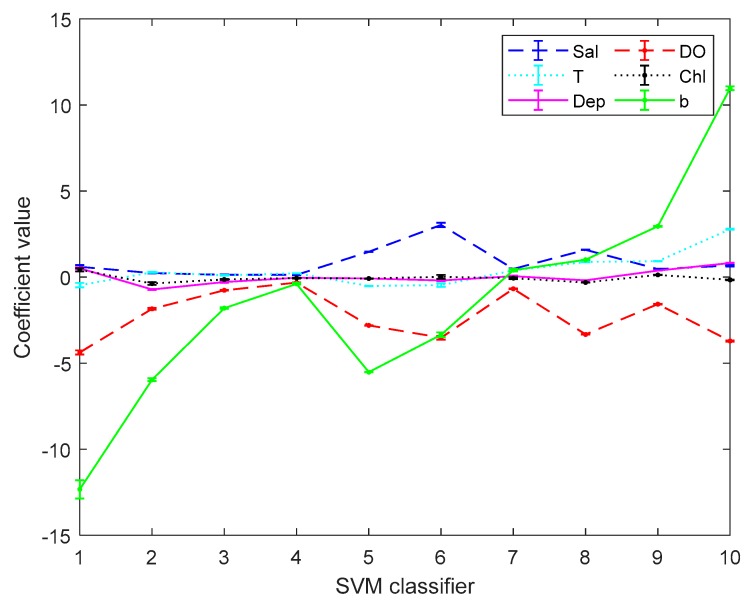
Coefficients by SVM after data preprocessing.

**Figure 7 sensors-19-03875-f007:**
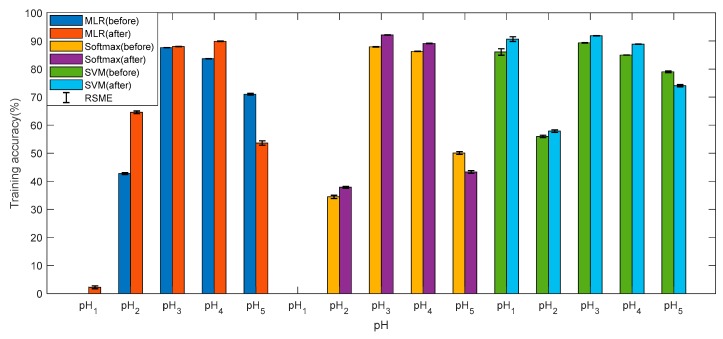
The training accuracy for each pH by three methods.

**Figure 8 sensors-19-03875-f008:**
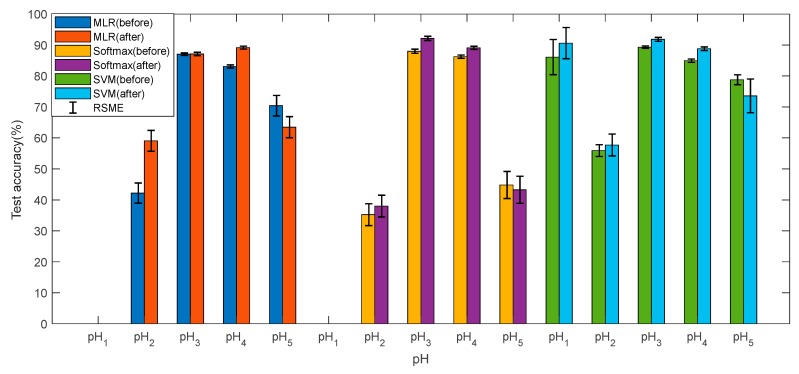
The test accuracy for each pH by three methods.

**Figure 9 sensors-19-03875-f009:**
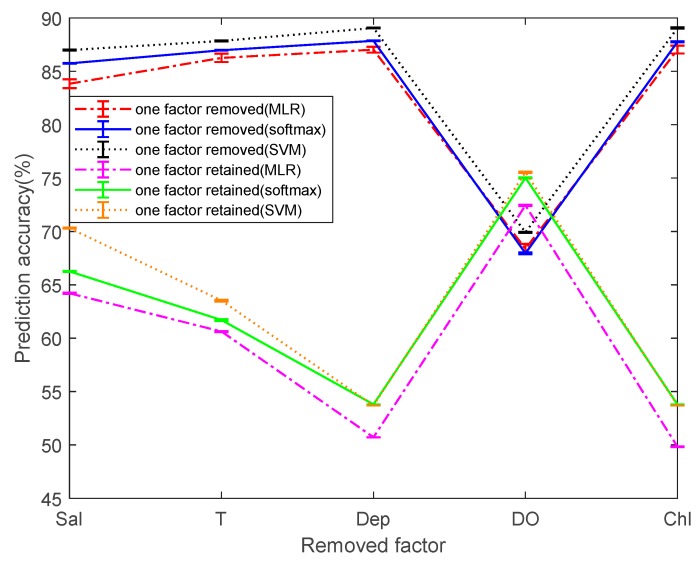
Prediction accuracy after one factor is removed or retained.

**Figure 10 sensors-19-03875-f010:**
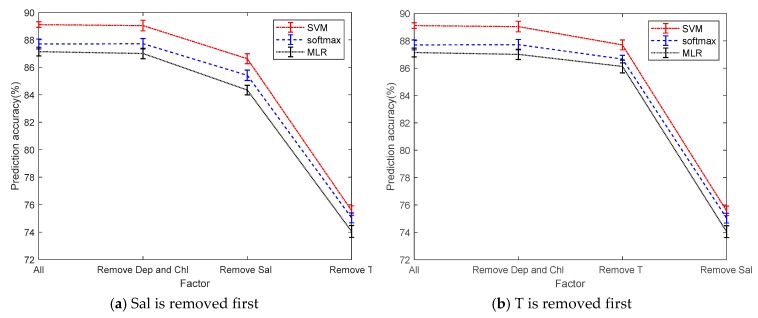
Prediction accuracy after simplifying the model.

**Table 1 sensors-19-03875-t001:** Coefficients and root mean squared error (RMSE) of multiple linear regression (MLR).

Variable	Parameters (Before)	RMSE	Parameters (After)	RMSE
**Intercept**	8.07438	3.32419 × 10^−5^	8.07572	3.07905 × 10^−5^
**Sal**	−0.02049	3.85279 × 10^−5^	−0.02002	3.41996 × 10^−5^
**T**	−0.00419	5.92817 × 10^−5^	−0.00354	3.72425 × 10^−5^
**Dep**	0.00100	4.43218 × 10^−5^	0.00121	3.28933 × 10^−5^
**DO**	0.03846	4.26921 × 10^−5^	0.03972	3.21011 × 10^−5^
**Chl**	0.00305	5.14602 × 10^−5^	0.00152	3.92131 × 10^−5^

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
