# Peer review of "Prediction of pH Value by Multi-Classification in the Weizhou Island Area"

_sensors, 2019, doi:10.3390/s19183875_

Round 1

Reviewer 1 Report

The work submitted is very interesting and correlates different sea water parameters with pH. Despites describing the methods employed accordingly, there are some issues concerning the discussion and conclusion of their work that must be clarified before considering its acceptance for publication in Sensors.

Authors conclude, based on Figure 10, that the highest prediction accuracy was obtained by the SVM model (89,04%). However, if one refers to Figure 10, the highest prediction accuracy is obtained with the MLR model (red line). A different behavior can be found in Figure 9, in which, SVM (black dotted line) has the highest prediction accuracy.

Also related to Figure 10, authors state in lines 291 and 292 that when parameter T is removed from the model the prediction accuracy changes little while after removing parameter Sal, there is a great change on prediction accuracy. However, in Figure 10 the opposite behavior can be noted, with a great change after removing the parameter T and a small change after removing Sal.

Also, authors state in lines 243 and 244 that the training and test accuracy of the MLR and Softmax regression for pH1 are almost zero. However, in Figures 7 and 8 this behavior is observed for MRL and SVM regression models, therefore, Softmax should be superior than the other two methods in predicting pH.

Can authors comment on that?

Some other issues that requires further clarifications are:

Line 75 – In the experiment, the data were recorded in 58 consecutive days? Which season (summer, spring…)? Line 77 – Could the authors provide some reference for the pH conversion formula? Line 199 – Could the authors explain the difference between cost function and real cost function? Why pH5 accuracy has decreased after data is preprocessed? Authors are encouraged to comment on that. Have the authors measured an unknown sample to validate their model?

Reviewer 2 Report

The authors studied the correlation between pH and various water quality parameters through three multi-classification approaches. It is an interesting and meaningful work, and shows a relatively high prediction accuracies in the investigated system, which may be beneficial for studying the ocean acidification issues. Following comments should be addressed before it can be accepted.

The accuracies of the sensors will affect the accuracies of the prediction models and the reliable and accurate data is the basis of the reliability of the model. The pH sensors show a relatively low accuracy, and only five different values can be measured, will this low accuracy influence the accuracies of the models? Authors provided the models on the correlations between pH and other quality parameters, and they are useful. It is suggested that authors explain the possible reasons why salinity, temperature and dissolved oxygen affect the pH, or what is the main reason causing these changes.

Reviewer 3 Report

Authors present here a full paper reporting the use some multi-classification approaches for prediction of pH values associated to ocean water quality.

In general, the authors dedicated most of the paper to describe the mathematical foundations of the tools they will use. These tools are widely known in the scientific world given the large number of applications where they can be used. In this sense there is no significant or relevant contribution in section 2.

Although the paper is well written and structured, the content related to specific data analysis and corresponding interpretation is very basic and lacking adequate scientific interpretation. At least, it is necessary to consider using additional statistical study apart of the basic cross-validation technique currently reported.

In particular, there are some relevant points that the authors should correct, improve or clarify:

Related to pre-processing stage (section 2.3). The procedure is not justified by the serious presence of interference measures but also not consider the way they were acquired signals. What is the origin of this interference? Is this related to the mentioned insufficient accuracy of the equipment used?.  It is necessary to include appropriate references and substantiate the process properly. How was established the abnormal value that is removed from records?. Data description in section 3 is not clear about dimensions. Table 1 should be edited in order to present numeric unified format. Figure 3 to Figure 6 are not significant enough to perceive variations or fluctuations of values including. Related to SVM. Which parameters were established to the employed model? What kernel function was used?

For all these reasons, I think that the paper is not acceptable for publication.

Round 2

Reviewer 1 Report

The revised version of the manuscript is now clear as to the text and legend that were dubious. The details that were provided by the authors improved the manuscript overall quality.

However, before its acceptance a minor detail must be corrected in line 76. I believe the author meant November 2018 and not 2019.

Reviewer 3 Report

Thank you for your updates. It became be a more strong.

Author Response

 Thank you very much for your invaluable suggestions.